# Qualitative Numerical Analysis of a Free-Boundary Diffusive Logistic Model

María Consuelo Casabán [1], Rafael Company [1,*], Vera N. Egorova [2] and Lucas Jódar [1]

1 Instituto de Matemática Multidisciplinar, Universitat Politècnica de València, Camino de Vera, s/n, 46022 Valencia, Spain
2 Depto de Matemática Aplicada y Ciencias de la Computación, Universidad de Cantabria, Avda. de los Castros, s/n, 39005 Santander, Spain
* Correspondence: rcompany@imm.upv.es

**Abstract:** A two-dimensional free-boundary diffusive logistic model with radial symmetry is considered. This model is used in various fields to describe the dynamics of spreading in different media: fire propagation, spreading of population or biological invasions. Due to the radial symmetry, the free boundary can be treated by a front-fixing approach resulting in a fixed-domain non-linear problem, which is solved by an explicit finite difference method. Qualitative numerical analysis establishes the stability, positivity and monotonicity conditions. Special attention is paid to the spreading–vanishing dichotomy and a numerical algorithm for the spreading–vanishing boundary is proposed. Theoretical statements are illustrated by numerical tests.

**Keywords:** free-boundary problem; diffusive logistic model; radial symmetry; spreading–vanishing dichotomy; numerical analysis; finite-difference method

**MSC:** 35R35; 65M06; 65M12; 65M22; 80A22

## 1. Introduction

Since the seminal works of Hotelling (1921) [1], Fisher (1937) [2] and Kolmogorov, Petrovski and Piskunov (1937) [3], time-dependent diffusion–reaction partial differential equations (PDE) have been commonly used for describing the growth and spread of populations and for propagation modelling in many different applications, such as biological invasions [4], epidemic spreading [5,6], wildfire propagation [7,8] or population genetics [9].

A crucial element of the propagation problems is the localisation of the spreading front. By the strong maximum principle for parabolic equations, the unknown population density $u(x,t)$ satisfies

$$u(x,t) > 0 \quad \forall x \in \mathbb{R}^n, \, t > 0, \tag{1}$$

although the initial population range is bounded.

In [10], Du and Lin introduced a modification of the Fisher-KPP diffusive logistic model by adding a Stefan's type condition for the boundary, such that the front has to be determined depending on time together with the unknown $u(x,t)$. Thus, the problem becomes a free boundary problem. The one-dimensional case is considered in [10], while the two-dimensional case with radial symmetry is studied in [11]. The cited works [10,11] have inspired many further studies for more complex models [6,12–14] and the model in a heterogeneous time-periodic environment [15].

One important advantage of considering this free boundary propagation model is the ability to capture the realistic dichotomy of two possible scenarios: spreading or vanishing. However, with the previous reaction–diffusion models without imposed free boundary, only spreading is observed. Some relevant theoretical results about the spreading–vanishing dichotomy are proved in [10,11].

The aim of this paper is to provide an efficient numerical method for the free boundary diffusive logistic model with radial symmetry. Since the best model may be wasted with a careless numerical treatment, a detailed numerical analysis is studied. Furthermore, reliable algorithms are proposed for numerical approximation of the relevant parameters for a spreading or vanishing scenario.

The aim of this paper is to present an effective numerical approach for solving the free boundary diffusive logistic model with radial symmetry based on front-fixing transformation. Given the potential loss of accuracy that can result from a poor numerical treatment, we provide a detailed numerical analysis. Additionally, the paper also includes a presentation of several significant theorems related to spreading–vanishing phenomena. Furthermore, we propose efficient algorithms for the numerical approximation of the relevant parameters for preading or vanishing scenario.

The paper is structured into several sections. Section 2, provides a concise explanation of the free boundary diffusive logistic model [11], followed by new results related to the spreading–vanishing dichotomy. In Section 3, we describe the methodology employed to derive numerical solutions for the unknown population density and the propagation front. This involved the development of a front-fixing method and the utilization of a finite difference scheme. Section 4 focuses on the numerical analysis of the solution, with specific emphasis on stability, positivity, and monotonicity. We then present numerical experiments to illustrate the results in Section 5. Finally, we conclude the paper in the last section.

## 2. Diffusive Logistic Model

In the present study we follow the approach introduced by Du and Lin [10] of using the free boundary formulation with the Stefan condition. In [11], authors extended this approach to a multidimensional case with radial symmetry. This assumption allows the reduction of the general multidimensional partial differential equation (PDE) to the one dimensional with the unique spatial variable $r = |x|$, where $x \in \mathbb{R}^N$. In this paper we consider $N = 2$, i.e., the two-dimensional problem. Hence, the population density of a spreading species $u(r, t)$ is found to be a positive solution of the following free-boundary PDE problem,

$$u_t(r,t) = D\Delta u(r,t) + u(r,t)(\alpha(r) - \beta(r)u(r,t)), \quad t > 0, \ 0 < r < H(t), \tag{2}$$

$$\frac{dH(t)}{dt} = -\mu u_r(H(t),t), \quad t > 0, \tag{3}$$

subject to the initial and boundary conditions

$$H(0) = H_0, \ u(r,0) = u_0(r), \quad 0 \le r \le H_0, \tag{4}$$

$$u_r(0,t) = 0, \quad u(H(t),t) = 0, \quad t > 0, \tag{5}$$

where $H(t)$ is the unknown moving boundary describing the spreading front of the species, $D$ is the diffusion rate, $\mu$ is a given positive proportionality constant between the moving boundary speed and the population gradient at the front, $\alpha(r)$ is the intrinsic growth rate and $\alpha(r)/\beta(r)$ is the habitat carrying capacity. Both $\alpha(r)$ and $\beta(r)$ are bounded functions, i.e., there are two positive constants $\kappa_1$ and $\kappa_2$, such that

$$\kappa_1 \le \alpha(r), \ \beta(r) \le \kappa_2, \quad \forall r \in [0, \infty). \tag{6}$$

Note that this model is radially symmetric but heterogeneous. The opportunity of considering these type of population dynamics models instead of the homogeneous ones is pointed out in [4,16,17].

Initial distribution $u_0(r)$ satisfies

$$u_0 \in C^2([0,H_0]), \quad u_0'(0) = u_0(H_0) = 0, \quad u_0(r) > 0 \ \forall r \in [0,H_0). \tag{7}$$

Due to the radial symmetry, $u(r, t)$ depends only on one spatial variable $r$ and Laplacian $\Delta u$ takes the form

$$\Delta u = u_{rr} + \frac{1}{r} u_r, \tag{8}$$

where the right side of equality (8) is the so-called Bessel operator [18,19].

The Stefan condition, expressed by Equation (3), ensures that the speed of the spreading front is directly proportional to the gradient of the population at the front. It is important to note that when $\mu = 0$, the free-boundary PDE problem is reduced to a fixed domain problem, which corresponds to the classical Fisher-KPP model [2].

*Spreading–Vanishing Dichotomy*

Previous theoretical studies [10,11] found that the population can either spread by occupying a new area or vanish without propagating depending on the initial front, population density and the parameter $\mu$. This phenomena is called a spreading–vanishing dichotomy. Summarising the results of Theorems 2.4, 2.5 and 2.10 of [11], we can formulate the following spreading criteria:

**Theorem 1** (Spreading–vanishing dichotomy [11]). *Let $(u(r, t), H(t))$ be the solution of the problem (2)–(5), then there is a positive constant $R^*$, such that the following situations are possible:*

1. *If $H_0 \geq R^*$, then the population spreads;*
2. *If $H_0 < R^*$, then there exists $\mu^* > 0$ depending on $u_0$ such that:*
   - *If $\mu > \mu^*$, then the population spreads;*
   - *If $\mu \leq \mu^*$, then the population vanishes.*

In the present paper, we focus on the computation of $R^*$ with the aim of providing a numerical complementation of the results in [11]. Following the ideas of [11], $R^*$ is established such that

$$\lambda_1(D, \alpha(r), R^*) = 1, \tag{9}$$

where $\lambda_1(D, \alpha(r), R^*)$ is the principal eigenvalue of the problem

$$\begin{cases} D\Delta\phi + \lambda\alpha(r)\phi = 0, & 0 < r < R^*, \\ \phi(R^*) = 0. \end{cases} \tag{10}$$

Hence, we aim to solve the inverse problem: for fixed $\lambda = 1$, find $R^*$ such that (10) holds. The following corollary defines $R^*$.

**Corollary 1** (Initial value problem for spreading–vanishing boundary). *Due to the radial symmetry, problem (10) for $\lambda = 1$, is equivalent to the following initial value problem (IVP):*

$$\begin{cases} D\Delta\phi + \alpha(r)\phi = 0, & r > 0, \\ \phi(0) = C, \\ \phi'(0) = 0, \end{cases} \tag{11}$$

*where $C > 0$ is some arbitrary constant. Then, $R^*$ is the first positive root of $\phi(r) = 0$.*

Note that the choice of the constant $C$ does not change the value $R^*$, which can be shown by the following lemma.

**Lemma 1.** *Let $\phi_1(r)$ and $\phi_2(r)$ be the solutions of IVP (11) for $C = C_1 > 0$ and $C = C_2 > 0$, respectively. If $r_0$ is a root of $\phi_1(r) = 0$, then $\phi_2(r_0) = 0$.*

**Proof of Lemma 1.** First, let us show that $\phi_2(r) = \kappa \phi_1(r)$, where $\kappa = \frac{C_2}{C_1}$. Since Equation (11) is linear, it is easy to see that

$$D\Delta(\kappa \phi_1(r)) + \alpha(r)(\kappa \phi_1(r)) = \kappa[D\Delta\phi_1(r) + \alpha(r)\phi_1(r)] = 0. \tag{12}$$

Moreover, $\phi_2(0) = \kappa \phi_1(0) = C_2$, and $\phi_2'(0) = \kappa \phi_1'(0) = 0$. Hence, $\phi_2(r)$ is a solution of IVP (11) with $C = C_2$.

Now, if $r_0$ is a root of $\phi_1(r) = 0$, then $\phi_2(r_0) = \kappa \phi_1(r_0) = 0$. □

Corollary 1 defines the algorithm for computation of $R^*$. The following theorem gives analytical expression for $R^*$ for the simplest case $\alpha = const$.

**Theorem 2.** *Spreading–vanishing boundary $R^*$ for the problem (2)–(5) with constant intrinsic growth rate $\alpha = const$ is defined as follows*

$$R^* = r_0 \sqrt{\frac{D}{\alpha}}, \tag{13}$$

*where $r_0$ is the first positive root of the Bessel function of the first kind, $r_0 = 2.40483$.*

**Proof of Theorem 2.** Since $\alpha = const$, from (8), the differential equation in problem (11) can be written as the following second-order linear differential equation

$$xy'' + ay' + bxy = 0, \quad \text{with } a = 1, \ b = \frac{\alpha}{D}. \tag{14}$$

The general solution of (14) is given in terms of the Bessel functions ([20], p. 241, eq. 63), by

$$y(x) = x^{\frac{1-a}{2}}\left(C_1 J_\nu(\sqrt{b}x) + C_2 Y_\nu(\sqrt{b}x)\right), \quad \nu = \frac{1}{2}|1 - a|, \tag{15}$$

where $J_\nu(x)$ and $Y_\nu(x)$ are the Bessel functions of the first and second kind, respectively.

In the case of (11), $a = 1$ and $b = \frac{\alpha}{D}$, and consequently $\nu = 0$, which leads to the following general solution

$$y(x) = C_1 J_0\left(\sqrt{\frac{\alpha}{D}}x\right) + C_2 Y_0\left(\sqrt{\frac{\alpha}{D}}x\right). \tag{16}$$

From (11), $\phi(0) = C$, hence, since $Y_0(x)$ is singular in $x = 0$ [21], $C_2 = 0$ and $C_1 = C$. Moreover, $\frac{d}{dx}J_0(x = 0) = 0$ [21], which agrees with the second initial condition in (11). Finally, the solution of the IVP (11) is found as follows

$$\phi(r) = C \cdot J_0\left(\sqrt{\frac{\alpha}{D}}r\right). \tag{17}$$

The distribution of zeroes for the Bessel functions of the first kind on the real line is known [21], and the first positive root is $r_0 = 2.40483$, hence

$$\sqrt{\frac{\alpha}{D}}R^* = r_0, \tag{18}$$

which proves the statement of the Theorem. □

In the general case $\alpha = \alpha(r)$, numerical methods have to be employed, see Algorithm 1. In the present paper, the numerical solution to the IVP (11) is obtained through the use of the Runge–Kutta–Fehlberg (RKF) method [22], an adaptive step-size method that estimates local errors and adjusts step size accordingly. By using fourth and fifth-order Runge–Kutta formulae, the RKF method can provide highly accurate solutions to IVPs. While factors such as initial conditions may affect the accuracy of the RKF method, it has been demonstrated

that the choice of $C$ does not impact the value of the first positive root. Given the high accuracy of the RKF method, root-finding algorithms are likely to converge quickly and accurately to the first positive root. If necessary, the numerical solution can be interpolated using computationally efficient and accurate methods, such as cubic spline interpolation, to obtain values at points of interest.

---

**Algorithm 1:** Computation of $R^*$.

---

**Data:** $D, \alpha(r)$
**Result:** $R^*$
$C \leftarrow 1$;
Solve IVP numerically:
$$\begin{cases} u_1' = u_2, \\ u_2' = -\frac{1}{r}u_2 - \frac{\alpha(r)}{D}u_1, \\ u_1(0) = C, \\ u_2(0) = 0. \end{cases}$$
$f(x) \leftarrow \text{interpolate}(u, x)$;
Solve $f(x) = 0$;
$R^* \leftarrow$ first positive root of $f(x)$

---

Once $R^*$ is found, Theorem 1 is applied. If $H_0 < R^*$ is chosen, we search for $\mu^*$ by using the simplified bisection method, see Algorithm 2.

---

**Algorithm 2:** Computation of $\mu^*$.

---

**Data:** $H_0, \alpha(r), \beta(r), D, u_0, tol$ ;                                    /* $H_0 < R^*$ */
**Result:** $\mu^*$
$\mu_1 \leftarrow 0, \mu_2 \leftarrow 10, N_{\max} \leftarrow 100$;
**for** $i = 1, 2$ **do**
    $u_i \leftarrow$ Solution of (2)–(5) for $\mu_i$ ;
    **if** $u_i'(r = 0, t) \leq 0 \ \forall t \in [0, T]$ **then**
        $f_i \leftarrow 0$;
    **else**
        $f_i \leftarrow 1$;
    **end**
**end**
$n \leftarrow 0$;
**while** $|\mu_1 - \mu_2| > tol \ and \ n < N_{\max}$ **do**
    $\mu_c \leftarrow \frac{\mu_1 + \mu_2}{2}$;
    $u_c \leftarrow$ Solution of (2)–(5) for $\mu_c$ ;
    **if** $u_c'(r = 0, t) \leq 0 \ \forall t \in [0, T]$ **then**
        $f_c \leftarrow 0$;
    **else**
        $f_c \leftarrow 1$;
    **end**
    **if** $f_c = 1$ **then**
        $\mu_2 \leftarrow \mu_c$;
    **else**
        $\mu_1 \leftarrow \mu_c$;
    **end**
    $n \leftarrow n + 1$;
**end**
$\mu^* \leftarrow \mu_c$

---

## 3. Numerical Algorithm

Problem (2)–(4) is a free-boundary PDE problem to be solved numerically. One of the challenges related to the problem is the free boundary that can be treated by the front-fixing Landau transformation [23,24], as it is done for the one-dimensional case [25]. In this paper, we extend this approach to the two-dimensional case.

The front-fixing method has provided reliable results for free boundary models in other fields, such as in mathematical finance [26] or civil engineering [27].

### 3.1. Front-Fixing Transformation

Under the assumption of the radial symmetry, the diffusive logistic model is described by the free-boundary PDE (2) with one spatial variables; hence, the Landau transformation takes the form

$$z(r,t) = \frac{r}{H(t)}, \quad V(z,t) = u(r,t). \tag{19}$$

Under the transformation (19), the moving domain $r \in (0, H(t))$ becomes the fixed one $z \in (0,1)$ at any time moment $t > 0$. Let us introduce the following notation

$$a(z,t) = \alpha(z \cdot H(t)) = \alpha(r), \tag{20}$$
$$b(z,t) = \beta(z \cdot H(t)) = \beta(r), \tag{21}$$

with the same constraint

$$0 < \kappa_1 \le a(z,t), \ b(z,t) \le \kappa_2. \tag{22}$$

By denoting $G(t) = H^2(t)$ and substituting (19) into the problem (2)–(5), one obtains

$$G(t)V_t = DV_{zz} + \left(\frac{D}{z} + \frac{z}{2}G'(t)\right)V_z + G(t)V(a(z,t) - b(z,t)V), \quad t > 0, \ 0 < z < 1, \tag{23}$$

$$G'(t) = -2\mu V_z(1,t), \quad t > 0, \tag{24}$$

subject to the initial and boundary conditions

$$G(0) = H_0^2, \ V(z,0) = u_0(zH_0), \quad 0 \le z \le 1, \tag{25}$$
$$V_z(0,t) = 0, \quad V(1,t) = 0, \quad t > 0, \tag{26}$$

Non-linear PDE problem (23)–(26) is solved numerically by using the explicit finite difference method (FDM) as described below.

### 3.2. Explicit FDM

First, let us introduce a bounded computational domain $\Omega = [0, T] \times [0, 1]$. We define the uniform grid

$$z_j = jh, \ h = 1/M, \quad t^n = nk, \ k = T/N, \tag{27}$$

where $N$ and $M$ are two given integer numbers. Then, by denoting

$$v_j^n \approx V(z_j, t^n), \quad g^n \approx G(t^n), \quad a_j^n = a(z_j, t^n), \quad b_j^n = b(z_j, t^n), \tag{28}$$

a forward in time central in space FD scheme for (23) is given as

$$g^n \frac{v_j^{n+1} - v_j^n}{k} = D\frac{v_{j+1}^n - 2v_j^n + v_{j-1}^n}{h^2} + \left(\frac{D}{z_j} + \frac{z_j}{2}\frac{g^{n+1} - g^n}{k}\right)\frac{v_{j+1}^n - v_{j-1}^n}{2h}$$

$$+ g^n v_j^n\left(a_j^n - b_j^n v_j^n\right), \quad 1 \le j \le M-1, \ 0 \le n \le N-1. \tag{29}$$

By reordering the terms in the scheme (29), one obtains

$$v_j^{n+1} = A_j^n v_{j-1}^n + B_j^n v_j^n + C_j^n v_{j+1}^n, \tag{30}$$

where

$$A_j^n = \frac{Dk}{h^2 g^n} - \frac{Dk}{2hg^n z_j} - \frac{z_j}{4h}\left(\frac{g^{n+1}}{g^n} - 1\right),$$ (31)

$$B_j^n = 1 - \frac{2Dk}{h^2 g^n} + k\left(a_j^n - b_j^n v_j^n\right),$$ (32)

$$C_j^n = \frac{Dk}{h^2 g^n} + \frac{Dk}{2hg^n z_j} + \frac{z_j}{4h}\left(\frac{g^{n+1}}{g^n} - 1\right).$$ (33)

The Stefan condition (24) is discretized by using the forward in time one-sided in space FD scheme as follows

$$\frac{g^{n+1} - g^n}{k} = -\mu\frac{3v_M^n - 4v_{M-1}^n + v_{M-2}^n}{h}.$$ (34)

The boundary conditions (26) are discretized as

$$V_z(0, t^n) \approx \frac{-3v_0^n + 4v_1^n - v_2^n}{2h} = 0, \quad v_M^n = 0.$$ (35)

Finally, the explicit FDM for problem (23)–(26) is written as follows

$$\begin{cases} g^{n+1} = g^n + \frac{k}{h}\mu\left(4v_{M-1}^n - v_{M-2}^n\right), \\ v_j^{n+1} = A_j^n v_{j-1}^n + B_j^n v_j^n + C_j^n v_{j+1}^n, \quad 1 \le j \le M - 1, \\ v_0^{n+1} = \frac{4}{3}v_1^{n+1} - \frac{1}{3}v_2^{n+1}, \\ v_M^{n+1} = 0, \quad 0 \le n \le N - 1, \end{cases}$$ (36)

together with the initial conditions $g^0 = H_0^2$ and $v_j^0 = u_0(z_j H_0)$, $0 \le j \le M$.

## 4. Numerical Analysis

In this section, we study the most important characteristics of the proposed numerical scheme, such as consistency and stability. Moreover, this section is dedicated to showing that the proposed numerical algorithm preserves crucial qualitative properties of the exact solution of the problem [10], such as positivity and monotonicity.

Consistency of the proposed numerical scheme can be easily established by following Section 3 of [25].

### 4.1. Positivity

Dealing with the models for spreading media, such as fire propagation, population spreading, etc., it is important to preserve non-negativity, otherwise the numerical solution is meaningless. Hence, in this subsection, we show that the proposed scheme (36) provides non-negative solutions under some conditions on the step size discretization.

**Theorem 3.** *With previous notation, $\{g^n\}_{0 \le n \le N}$ is positive increasing in a time sequence.*

**Proof of Theorem 3.** Let us use the induction principle on the index $n$. For $n = 0$, from the initial conditions (4), it follows that

$$\frac{du_0}{dr}(H_0) < 0,$$ (37)

which becomes $v_z(1, 0) < 0$ under the front-fixing transformation.

By assuming that $v(z, t)$ is at least twice differentiable with respect to $z$ and by using the Taylor's expansion about $(z_M, 0)$, $z_M = 1$, one obtains

$$v^0_{M-1} = v^0_M - hv_z(1, 0) + \mathcal{O}(h^2), \tag{38}$$
$$v^0_{M-2} = v^0_M - 2hv_z(1, 0) + \mathcal{O}(h^2). \tag{39}$$

Since $v^0_M = 0$ and $v_z(1, 0) < 0$,

$$\Delta^0 = 4v^0_{M-1} - v^0_{M-2} = -2hv_z(1, 0) + \mathcal{O}(h^2) > 0, \tag{40}$$

for small enough values of $h$.

Hence, from (36), the positivity of $g^1$ is established for any $\mu > 0$

$$g^1 = g^0 + \frac{k}{h}\mu\Delta^0 > g_0 > 0. \tag{41}$$

Hence, the base case $n = 0$ is proven. Now, assuming that the statement of the Theorem holds true up to some integer $n$, let us prove that it holds for $(n + 1)$ as well.

Let us denote

$$\Delta^n = 4v^n_{M-1} - v^n_{M-2}, \quad n \geq 0. \tag{42}$$

Under the induction hypothesis, $g^n > g^{n-1}$, then for sufficiently small $k$, $\Delta^{n-1} > 0$ from the first equation of (36). As

$$\Delta^n = \Delta^{n-1} + \mathcal{O}(k), \tag{43}$$

it follows that $\Delta^n > 0$ for small enough values of $k$ and, consequently, $g^{n+1} > g^n$. Then, the statement of the Theorem is straightforward:

$$g^{n+1} > g^n > \ldots > g^1 > g^0 > 0. \tag{44}$$

□

The next lemmas show the positivity of the coefficients defined in (31)–(33), which implies directly the positivity of the solution.

**Lemma 2.** *Coefficients $A^n_j$ and $C^n_j$, defined by (31) and (33), respectively, are positive.*

**Proof of Lemma 2.** From Theorem 3 and definition (33), $C^n_j > 0$, as it is a sum of positive terms.

Since $h \leq z_j < Mh = 1$, for any $j = 1, \ldots, M - 1$, from (31), the following estimation is obtained

$$A^n_j \geq \frac{Dk}{h^2 g^n} - \frac{Dk}{2g^n h^2} - \frac{1}{4h}\left(\frac{g^{n+1}}{g^n} - 1\right) = \frac{k}{h^2 g^n}\left(\frac{D}{2} - \frac{h}{4k}(g^{n+1} - g^n)\right). \tag{45}$$

From (34) and (42), one gets

$$h\frac{g^{n+1} - g^n}{k} = \mu\Delta^n. \tag{46}$$

By using the Taylor's expansion about $(z_M, t^n)$ with sufficiently small $h$,

$$\Delta^n = -2hv_z(1, t^n) + \mathcal{O}(h^2). \tag{47}$$

Hence, $h\frac{g^{n+1} - g^n}{k} = \mathcal{O}(h)$, which together with (45) implies the positivity of $A^n_j$ for sufficiently small $h$. □

In order to show the positivity of $B_j^n > 0$, $1 \leq j \leq M - 1$, and $0 < v_j^n \leq M_0$, $0 \leq j \leq M - 1$, we consider two possible cases related to the relation between the initial population density $u_0(r)$ and the carrying capacity of the species $\frac{\alpha(r)}{\beta(r)}$:

1. $0 < u_0(r) < C_0$, $0 \leq r \leq H_0$,
2. $\max_{0 \leq r \leq H_0} \{u_0(r)\} = M_0 \geq C_0$,

where $C_0 = \sup \left\{ \frac{\alpha(r)}{\beta(r)}, \ 0 \leq r < \infty \right\}$.

Let us introduce the following notation

$$0 < \kappa_1 \leq \alpha_1 \leq \alpha(r) \leq \alpha_2 \leq \kappa_2, \quad 0 \leq r < \infty, \tag{48}$$

$$0 < \kappa_1 \leq \beta_1 \leq \beta(r) \leq \beta_2 \leq \kappa_2, \quad 0 \leq r < \infty, \tag{49}$$

then, $C_0$ is bounded as follows

$$\frac{\alpha_1}{\beta_2} \leq C_0 \leq \frac{\alpha_2}{\beta_1}. \tag{50}$$

Let us consider the first case, $0 < u_0(r) < C_0$, $0 \leq r \leq H_0$. We use the induction principle on the index $n$ to prove that under a condition to be found, the coefficients $B_j^n$ are positive and also the numerical solution is positive and upper bounded by $C_0$, $0 < v_j^n < C_0$, $0 \leq j \leq M - 1$, $0 \leq n \leq N$.

For $n = 0$, since $0 < u_0(r) < C_0$, then from the initial conditions (25), $0 < v_j^0 < C_0$, $0 \leq j \leq M - 1$. Moreover,

$$\begin{aligned}
B_j^0 &= 1 + k\left(a_j^0 - b_j^0 v_j^0\right) - \frac{2Dk}{h^2 g^0} > 1 + k\left(\alpha_1 - \beta_2 \frac{\alpha_2}{\beta_1}\right) - \frac{2Dk}{h^2 g^0} \\
&= 1 - k\left[\left(\frac{\alpha_2 \beta_2}{\alpha_1 \beta_1} - 1\right)\alpha_1 + \frac{2D}{h^2 g^0}\right] > 0, \quad 1 \leq j \leq M - 1,
\end{aligned} \tag{51}$$

under the following constrain

$$k < Q_1 h^2, \quad Q_1 = \frac{g^0}{2D + h^2 \alpha_1 g^0 \left(\frac{\alpha_2 \beta_2}{\alpha_1 \beta_1} - 1\right)}. \tag{52}$$

Now, let us assume that for some fixed $n$, the following is fulfilled

$$B_j^n > 0, \ 1 \leq j \leq M - 1, \qquad 0 < v_j^n < C_0, \ 0 \leq j \leq M - 1. \tag{53}$$

Then, $v_j^{n+1}$ defined by (30), is positive due to the positivity of all coefficients $A_j^n$, $B_j^n$, $C_j^n$ and the solution at the $n$th time level. On the other hand, $v_j^{n+1}$ can be considered as a function of $v_j^n$, i.e., $v_j^{n+1} = f(v_j^n)$ for $1 \leq j \leq M - 1$. Then, the first derivative

$$\frac{\partial f}{\partial v_j^n} = B_j^n - k b_j^n v_j^n = 1 + k\left(a_j^n - 2b_j^n v_j^n - 2\frac{D}{h^2 g^n}\right) > 1 + k\left(\alpha_1 - 2\beta_2 \frac{\alpha_2}{\beta_1} - 2\frac{D}{h^2 g^0}\right) > 0, \tag{54}$$

if

$$k < Q_2 h^2, \quad Q_2 = \frac{g^0}{2D + h^2 \alpha_1 g^0 \left(\frac{2\alpha_2 \beta_2}{\alpha_1 \beta_1} - 1\right)} < Q_1. \tag{55}$$

The positivity of the first derivative $\frac{\partial f}{\partial v_j^n}$ means that $v_j^{n+1}$ is increasing with respect to $v_j^n$, $0 < v_j^n < C_0$. Hence,

$$v_j^{n+1} < (A_j^n + B_j^n + C_j^n)C_0 = \left(1 + k a_j^n \left(1 - \frac{C_0}{a_j^n / b_j^n}\right)\right) C_0 \leq C_0, \quad 1 \leq j \leq M - 1. \tag{56}$$

For $j = 0$, by applying the Taylor's expansion about $(0, t^{n+1})$, it is found that

$$v_0^{n+1} = v_1^{n+1} + \mathcal{O}(h),$$ (57)

which leads to the boundedness of $v_0^{n+1}$, and, consequently,

$$0 < v_j^{n+1} < C_0, \quad 0 \le j \le M - 1.$$ (58)

To finish the proof for the first case, we show that under the constrain (52), $B_j^{n+1} \ge 0$, as

$$B_j^{n+1} > 1 + k\left(\alpha_1 - \beta_2 \frac{\alpha_2}{\beta_1} - \frac{2D}{h^2 g^0}\right) > 0.$$ (59)

Now, let us consider the second case, $\max_{0 \le r \le H_0}\{u_0(r)\} = M_0 \ge C_0$. Analogously, we use the induction principle.

For $n = 0$,

$$0 < v_j^0 = u_0(H_0 z_j) \le M_0, \quad 0 \le j \le M - 1.$$ (60)

By defining

$$C_m = \inf_{r \in \mathbb{R}^+}\left\{\frac{\alpha(r)}{\beta(r)}\right\}, \quad \frac{\alpha_1}{\beta_2} \le C_m \le C_0 \le M_0,$$ (61)

one obtains

$$a_j^0 - b_j^0 v_j^0 \ge b_j^0\left(\frac{a_j^0}{b_j^0} - M_0\right) \ge b_j^0(C_m - M_0) \ge \beta_2(C_m - M_0).$$ (62)

Therefore,

$$B_j^0 \ge 1 - k\left(\beta_2(M_0 - C_m) + \frac{2D}{h^2 g^0}\right) > 0,$$ (63)

if $k$ satisfies the following condition

$$k < Q_3 h^2, \quad Q_3 = \frac{g^0}{2D + h^2 g^0 \beta_2(M_0 - C_m)}.$$ (64)

Let us assume that for some fixed $n$, $B_j^n > 0$, $1 \le j \le M - 1$, and $0 < v_j^n \le M_0$, $0 \le j \le M - 1$. To complete the proof we have to show that it holds for $n + 1$.

The positivity of $v_j^{n+1}$, $1 \le j \le M - 1$ is straightforward from the definition (30) and positivity of the coefficients at the time level $t^n$. By using the Taylor's expansion about $(z_0, t^{n+1})$, it is easy to show that $v_0^{n+1} > 0$.

Analogously to the first case, we consider $v_j^{n+1} = f(v_j^n)$ with the first partial derivative

$$\frac{\partial f}{\partial v_j^n} = 1 + k\left(a_j^n - 2b_j^n v_j^n - 2\frac{D}{h^2 g^n}\right) \ge 1 + k\left(\beta_2(C_m - 2M_0) - \frac{2D}{h^2 g^0}\right) > 0,$$ (65)

if

$$k < Q_4 h^2, \quad Q_4 = \frac{g^0}{2D + h^2 g^0 \beta_2(2M_0 - C_m)} < Q_3.$$ (66)

Since $0 < v_j^n \le M_0$, $1 \le j \le M - 1$, by the hypothesis of the induction, then by taking into account that $\frac{1}{C_0} = \inf_{r \in \mathbb{R}^+}\left\{\frac{\beta(r)}{\alpha(r)}\right\}$, one obtains

$$v_j^{n+1} \le f(M_0) \le (A_j^n + B_j^n + C_j^n)M_0 \le \left(1 + ka_j^n\left(1 - \frac{M_0}{C_0}\right)\right)M_0 \le M_0.$$ (67)

For $j = 0$, if $v_0^{n+1} > M_0$, then $\frac{\partial v}{\partial z}|_{z=0} > 0$, which contradicts the boundary conditions. Hence, $v_0^{n+1} \leq M_0$.

Finally, if $k$ satisfies (64), then

$$B_j^{n+1} \geq 1 + k \left( \beta_2(C_m - M_0) - \frac{2D}{h^2 g^0} \right) > 0. \tag{68}$$

Summarising, the following Lemma and Theorem about the positivity condition are formulated.

**Lemma 3.** *Let us consider the numerical scheme* (36) *with sufficiently small step sizes h and k such that*

$$k < Qh^2, \tag{69}$$

*where* $Q = \min\{Q_2, Q_4\}$, *defined in* (55) *and* (66). *Then, coefficients* $B_j^n$, $1 \leq j \leq M - 1$, $0 \leq n \leq N - 1$, *defined by* (32) *are positive.*

**Theorem 4** (Positivity condition). *The numerical solution computed by the scheme* (36) *with sufficiently small step sizes h and k such that*

$$k < Qh^2, \quad Q = \min\{Q_2, Q_4\} \tag{70}$$

*where* $Q_2$, $Q_4$ *are defined in* (55), *and* (66) *is positive and bounded:*

$$0 \leq v_j^n \leq P_0, \quad 0 \leq j \leq M, \ 0 \leq n \leq N, \tag{71}$$

*where* $P_0 = \max\{M_0, C_0\}$.

*4.2. Stability*

In the present study, we use the definition of stability proposed in [28], p. 92, based on the supremum norm of a vector. Therefore, we start this subsection with recalling the definition of the stability.

**Definition 1** (Stability). *The numerical scheme* (36) *is said to be* $\| \cdot \|_\infty$*-stable in the domain* $[0, 1] \times [0, T]$, *if for every partition with step sizes* $h = \frac{1}{M}$ *and* $k = \frac{T}{N}$, *the following holds*

$$\|v^n\|_\infty \leq K, \quad 0 \leq n \leq N, \tag{72}$$

*where* $v^n = [v_0^n, v_1^n, \ldots, v_M^n]^T$ *is the vector solution of the scheme* (36) *and* $K > 0$ *is some constant independent of n and the step sizes h and k.*

By using this definition, from Theorem 4, one can take $K = P_0 = \max\{M_0, C_0\}$, where $M_0 = \max\{u_0(r) : 0 \leq r \leq H_0\}$ and $C_0 = \sup\left\{ \frac{\alpha(r)}{\beta(r)}, \ 0 \leq r < \infty \right\}$. As $K$ is independent of $n$ and the step sizes $h$ and $k$, the following result has been established.

**Theorem 5** (Stability). *With the previous notation, for small enough step sizes h and k satisfying the positivity condition* (70), *the scheme* (36) *is* $\| \cdot \|_\infty$*-stable in the domain* $[0, 1] \times [0, T]$.

*4.3. Monotonicity*

Apart from the basic qualitative properties, such as stability and positivity, we study the monotonicity of the numerical solution.

**Theorem 6** (Monotonicity). *Let* (36) *be a numerical scheme for the problem* (23)–(25), *where* $\alpha(r)$ *is a monotone decreasing function and* $\beta(r)$ *is a monotone increasing function. If h is sufficiently small and k satisfies the following constraint*

$$k < \tilde{Q}h^2, \tag{73}$$

*where* $\tilde{Q} > 0$ *is some positive constant that can be found in terms of the parameters of the problem, then the scheme* (36) *preserved the monotonicity of the numerical solution,*

$$v_j^n \geq v_{j+1}^n, \quad \Rightarrow \quad v_j^{n+1} \geq v_{j+1}^{n+1}, \quad 0 \leq j \leq M - 1. \tag{74}$$

**Proof of Theorem 6.** From (30),

$$
\begin{aligned}
v_j^{n+1} - v_{j+1}^{n+1} &= (v_j^{n+1} - v_j^n) + (v_j^n - v_{j+1}^n) - (v_{j+1}^{n+1} - v_{j+1}^n) \\
&= A_j^n v_{j-1}^n + k\left(a_j^n - b_j^n v_j^n - \frac{2D}{h^2 g^n}\right)v_j^n + C_j^n v_{j+1}^n \\
&\quad + (v_j^n - v_{j+1}^n) - \left(A_{j+1}^n v_j^n + k\left(a_{j+1}^n - b_{j+1}^n v_{j+1}^n - \frac{2D}{h^2 g^n}\right)v_{j+1}^n\right) - C_{j+1}^n v_{j+2}^n \\
&\geq (v_j^n - v_{j+1}^n) + (A_j^n - A_{j+1}^n)v_j^n + (C_j^n - C_{j+1}^n)v_{j+1}^n \\
&\quad - k\frac{2D}{h^2 g^n}(v_j^n - v_{j+1}^n) + k\left((a_j^n - b_j^n v_j^n)v_j^n - (a_{j+1}^n - b_{j+1}^n)v_{j+1}^n)v_{j+1}^n\right) \\
&\geq (v_j^n - v_{j+1}^n)\left(1 - k\left(\frac{D}{4h^2 g^n} + \frac{2D}{h^2 g^n}\right) + k b_{j+1}^n (v_{j+1}^n + v_j^n)(v_{j+1}^n - v_j^n)\right) \\
&\geq (v_j^n - v_{j+1}^n)\left(1 - k\left(\frac{9D}{4h^2 g^0} + 2P_0\beta_2\right)\right) \geq 0,
\end{aligned}
\tag{75}
$$

*where* $P_0 = \max\{M_0, C_0\}$, *if*

$$k \leq \tilde{Q}h^2, \quad \tilde{Q} = \frac{4g^0}{9D + 8h^2 g^0 P_0 \beta_2}. \tag{76}$$

□

## 5. Numerical Results

### 5.1. Constant Intrinsic Growth Rate and Carrying Capacity

First, we consider the case $\alpha = const$, $\beta = const$. It has been shown in Section 2 that in this case, $R^*$ can be found by applying Theorem 2. In the following example, we use the proposed numerical algorithm to approximate $R^*$.

**Example 1.** *In the logistic model* (2)–(5), *we set the following default parameters*

$$D = 1, \ \mu = 2, \ \alpha = \beta = 1, \ H_0 = 3, \ u_0(r) = 1 - \left(\frac{r}{H_0}\right)^2. \tag{77}$$

In order to calculate $R^*$, we solve numerically the IVP

$$D\phi'' + \frac{D}{r}\phi' + \alpha\phi = 0, \quad \phi(0) = 1, \ \phi'(0) = 0, \tag{78}$$

by using the Runge–Kutta–Fehlberg method, and then the first positive root is calculated by using the secant method. The found value $R^* = 2.4056$ agrees with the theoretical value $R_t^* = r_0\sqrt{\frac{D}{\alpha}} = 2.4048$. Since the default value $H_0 = 3 > R^*$, spreading is observed, as well as for $H_0 = 2.5$ and $H_0 = 2$. However, $H_0 = 1$ leads to vanishing, as shown in Figure 1, where the free boundary $H(t)$ is increasing in time for $H_0 = 2, 2.5, 3$ and constant for $H_0 = 1$.

Despite $H_0 = 2 < R^*$, the population still spreads, see Figure 1. It is explained by Theorem 1, case $\mu > \mu^*$. The threshold value $\mu^*$ is calculated by Algorithm 2, and it is found that $\mu^* = 0.2682$. Since the default value is $\mu = 2 > \mu^*$, the population spreads for $H_0 = 2$.

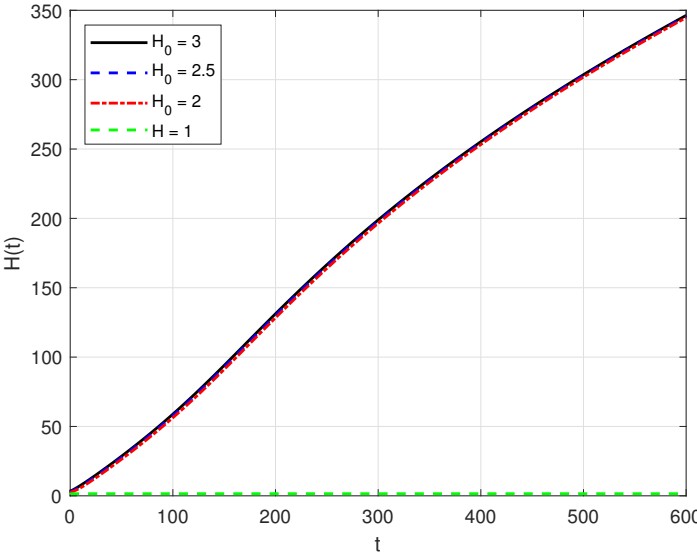

**Figure 1.** Free boundary $H(t)$ for the logistic model (2)–(5) with parameters given in (77) and various $H_0$.

The spreading–vanishing dichotomy for $H_0 = 2$ is illustrated in Figure 2. For the simulations we set $H_0 = 2 < R^*$ and two values for $\mu$: one was $\mu^*$, and the other was 10% higher, i.e., $\mu = 1.1\mu^*$.

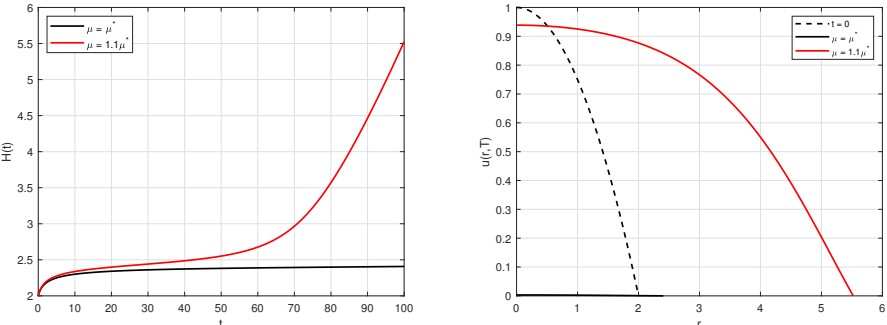

**Figure 2.** Numerical solution for Example 1: Free boundary $H(t)$ (**left**) and population distribution at the final moment $u(r, T)$ (**right**) for $H_0 = 2$ and $\mu = \mu^*$ (black solid line) and $\mu = 0.29507 > \mu^*$ (red solid line), which correspond to vanishing and spreading, respectively. The dashed line in the right plot is the initial population distribution $u_0(r)$.

Note that $T$ should be chosen properly and large enough since the population may decrease, but after some time, it starts recovering and finally, grows. It is illustrated by the following example.

**Example 2.** *In the logistic model (2)–(5), we set the following default parameters*

$$D = 1, \ \mu = 0.27, \ \alpha = \beta = 1, \ H_0 = 2, \ u_0(r) = 1 - \left(\frac{r}{H_0}\right)^2. \tag{79}$$

As shown in the previous example, $H_0 = 2 < R^*$ and $\mu = 0.27 > \mu^* = 0.2682$. We set $T = 200$ and plot the maximum population $u_{\max}(t)$, which in accordance with

Theorem 6 about the monotonicity preserving property of the numerical solution, is $u(0, t)$, see Figure 3.

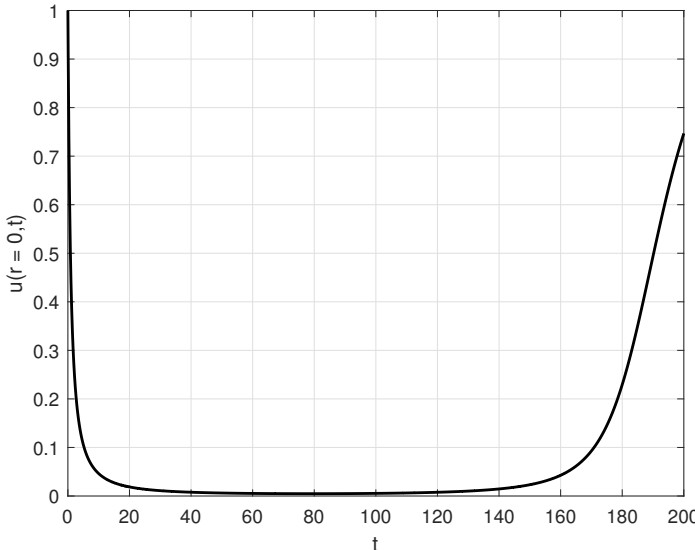

**Figure 3.** Maximum population with respect to time for the parameters (79) in Example 2.

If one considers the time horizon until $T = 100$, the population can be considered as vanished; however, if the time horizon is enlarged, until $T = 200$, the recovering effect is observed.

*5.2. General Case $\alpha(r)$, $\beta(r)$*

Now, let us illustrate the theoretical results established above with numerical examples for the general case $\alpha(r)$, $\beta(r)$.

**Example 3.** *In the logistic model* (2)–(5), *we set the following default parameters*

$$D = 1, \ \mu = 2, \ \alpha(r) = \frac{2r+3}{2r+2}, \ \beta(r) = \frac{2r+1}{2r+2}, \ H_0 = 3, \ u_0(r) = 1 - \left( \frac{r}{H_0} \right)^2, \ T = 3. \quad (80)$$

Analogous to the previous subsection, we start with the spreading–vanishing dichotomy. By Algorithm 1, we calculate $R^* = 2.1289$, which is lower than in the constant case. Hence, for $H_0 > R^*$, the population spreads, see Figure 4 (left plot).

Note that in the general case, the habitat carrying capacity varies between 1 and 3 (for $\alpha(r)$ and $\beta(r)$ defined by (80)). In this case, the initial population density $u_0(r)$ is less than the supremum of the carrying capacity $C_0$, and then the population can grow from the initial state but can not reach $C_0$.

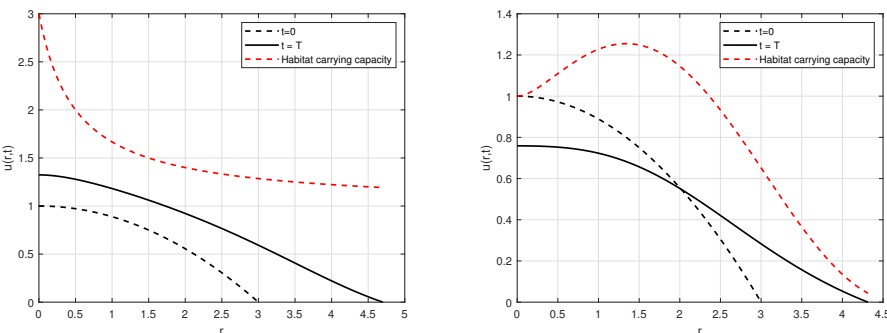

**Figure 4.** Population at $t = 0$ and $t = T$ for $\alpha(r) = \frac{2r+3}{2r+2}$ (**left**) and $\alpha(r) = 0.5 * (1 + \sin(r))$ (**right**).

Theorem 6 states that the proposed numerical method preserves the monotonicity of the solution with respect to $r$, if $\alpha(r)$ is monotone decreasing and $\beta(r)$ is monotone increasing functions. This property is also shown in Figure 4 (left plot); the solution is monotone non-increasing. However, if we change the intrinsic growth rate,

$$\alpha(r) = 0.5 \cdot (1 + \sin(r)), \tag{81}$$

then the monotonicity is lost, see the right plot of Figure 4.

In order to check the monotone behaviour of the solution, Figure 5 presents the slope $\Delta = \frac{du}{dr}$ at the moment $t = T$ for both monotone and osculating $\alpha(r)$. Note that for monotone $\alpha(r)$, the slope is negative; while for osculating $\alpha(r)$, the slope changes sign several times.

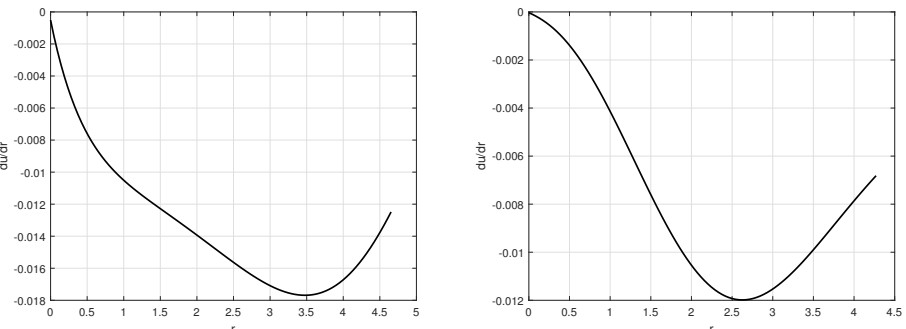

**Figure 5.** Slope at $t = T$ for $\alpha(r) = \frac{2r+3}{2r+2}$ (**left**) and $\alpha(r) = 0.5 * (1 + \sin(r))$ (**right**).

Finally, we check the stability condition proposed in Theorem 4. For that purpose, we consider the problem (2)–(5) with $T = 0.1$. We set $M = 100$, so $h = 10^{-2}$. By using the condition of Theorem 4 and (76), we find

$$Q = \min\{Q_2, Q_4, \tilde{Q}\} = \min\{4.4899, 4.4979, 3.9968\} = 3.9968. \tag{82}$$

In Figure 6, we compare the free boundary and the solution at $r = 0$ calculated by the proposed algorithm with $k_1 = Qh^2$ and $k_2 = 1.25Qh^2$.

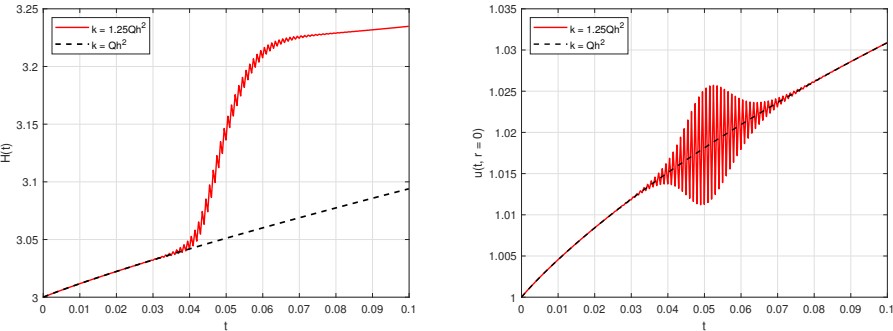

**Figure 6.** Stable (black dashed line) and unstable (red solid line) solution of the problem (2)–(5) with parameters (80), $T = 0.1$: free boundary $H(t)$ (**left** plot) and maximum population $u(r = 0, t)$ (**right** plot).

To complete the study, let us consider a case when the initial population exceeds the carrying capacity, i.e., the model with parameters (80), but with the initial population defined as follows

$$u_0(r) = 4 - \left(\frac{2r}{H_0}\right)^2. \tag{83}$$

The numerical solution at various time moments $t = 0, 0.5, 3, 10$, is plotted in Figure 7. Note that the initial population distribution is higher than the carrying capacity. At the

first time moment $t = 0.5$, the population is still above the carrying capacity, but recession is observable compared with the initial value. At $t = 3$, the population curve is below the carrying capacity line. Since the parameters considered in this example guarantee spreading behaviour, we observe that at $t = 10$, the population grows up to the carrying capacity. Hence, this example shows that the model and proposed numerical algorithm adapt to any initial conditions and preserve theoretical properties of the population.

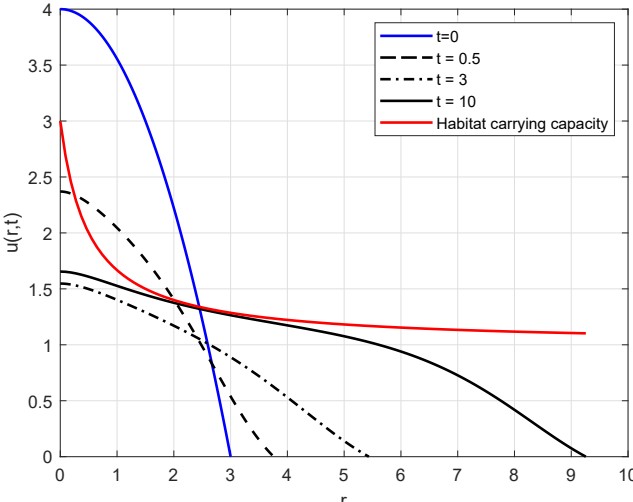

**Figure 7.** Numerical solution for Example 3 with parameters (80) and initial condition (83) at various moments $t$.

The proposed numerical algorithm is compared with known numerical methods for solving free boundary time-dependent PDE problems, such as the Level Set method (LSM) [29]. In the case of the PDE (2), it is the 1D version of the LSM. The key idea behind the level set method is to represent the interface or shape is the zero level set of a higher-dimensional function. In other words, we embed the interface or shape in a higher-dimensional space and define a scalar function that is negative inside the population and positive outside it. The zero level set of this function represents the front or boundary. By evolving this function over time, we can track the motion of the interface. The evolution of the function is solved using explicit FDM.

One of the advantages of the level set method is its natural extension to higher dimensions and the fact that it can handle topological changes, such as merging or splitting of interfaces, which are challenging for other numerical methods. The level set method is useful for problems involving complex geometry or multiple interacting interfaces. However, for the problem (2) with radial symmetry, this characteristic is not urgent.

Table 1 displays the root-mean-square error between the results obtained from the front-fixing (FF-FDM) method and LSM, as well as the corresponding computational times for different numbers of spatial grid points. To ensure parity, we preserve the parabolic mesh ratio and keep the temporal step-size constant for each $M$ value in both methods. Figure 8 shows the numerical solutions obtained from both methods. The small discrepancy near the free boundary observed in the results can be attributed to the approach used by the FF-FDM method, which calculates the free boundary as part of the solution. In contrast, the LSM method simply determines whether the solution lies inside or outside the population domain. This subtle difference in methodology can affect the accuracy of the solution near the free boundary, and should be taken into consideration when selecting an appropriate numerical method for solving similar problems.

**Table 1.** Root-mean-square error between the numerical solutions $u(r, T)$ obtained from the front-fixing (FF-FDM) method and LSM and corresponding CPU-time in seconds for both methods.

| *M* (Spatial Discretization) | LSM CPU-Time, s | FF-FDM CPU-Time, s | RMSE |
|---|---|---|---|
| 100 | 0.0312 | 0.0012 | 0.0079 |
| 200 | 0.0313 | 0.0937 | 0.0142 |
| 400 | 0.4687 | 0.5312 | 0.0150 |
| 800 | 2.2656 | 2.7500 | 0.0129 |
| 1600 | 16.3750 | 17.8125 | 0.0100 |
| 3200 | 148.5000 | 148.0625 | 0.0073 |

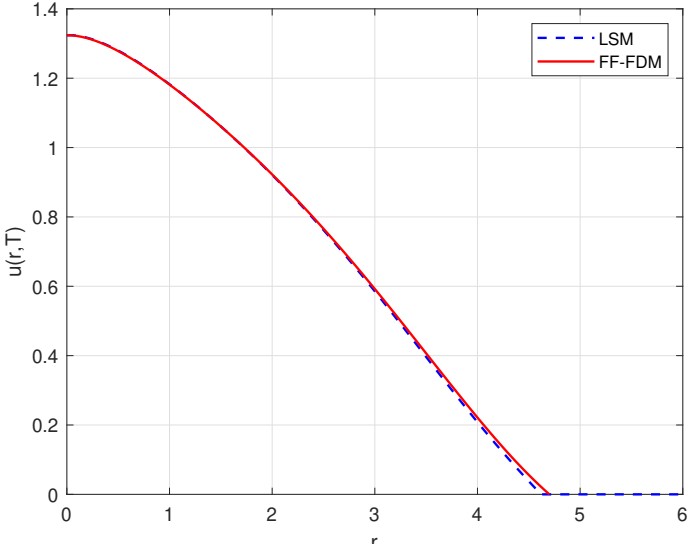

**Figure 8.** Numerical solutions $u(r, T)$ for $T = 3$ obtained from the front-fixing finite difference method (FF-FDM) and the level set method (LSM) for the problem in Example 3.

## 6. Conclusions

The current paper presents a novel and highly efficient numerical algorithm for solving a two-dimensional diffusive logistic model with radial symmetry. The proposed front-fixing method is found to be more robust and easier to implement compared to the widely-used level set method. Moreover, the numerical analysis shows that the proposed method is conditionally stable, consistent with the original PDE problem, and preserves the positivity and monotonicity of the solution.

Furthermore, the paper presents several important theorems that have been proven, and a numerical algorithm for the spreading–vanishing boundary for the general case of non-constant parameters $\alpha(r)$ and $\beta(r)$. All theoretical statements have been illustrated by numerical examples, which demonstrate the crucial role of the proposed stability conditions and the importance of a large enough time horizon for analyzing the spreading–vanishing dichotomy.

Overall, the proposed algorithm provides a powerful tool for solving diffusive logistic models with radial symmetry, and the theoretical analysis presented in the paper offers valuable insights into the dynamics of such models. The results and methods presented in the paper may have significant implications for a range of fields, including ecology, population dynamics, and epidemiology.

**Author Contributions:** Conceptualization, M.C.C., R.C., V.N.E. and L.J.; Methodology, M.C.C., R.C., V.N.E. and L.J.; Validation, M.C.C., R.C., V.N.E. and L.J.; Formal analysis, M.C.C., R.C., V.N.E. and L.J.; Investigation, M.C.C., R.C., V.N.E. and L.J.; Writing—review & editing, M.C.C., R.C., V.N.E. and L.J. All authors have read and agreed to the published version of the manuscript.

**Funding:** This research was partially funded by the Spanish Ministry of Science, Innovation and Universities, State Research Agency, National Research and Development Plan 2019 grant number PID2019-107685RB-I00.

**Data Availability Statement:** Data sharing not applicable.

**Conflicts of Interest:** The authors declare no conflict of interest.

## Abbreviations

The following abbreviations are used in this manuscript:

| | |
|---|---|
| FD | Finite difference |
| FDM | Finite difference method |
| FF | Front-fixing |
| IVP | Initial value problem |
| LSM | Level set method |
| PDE | Partial differential equation |
| RKF | Runge–Kutta–Fehlberg |
| RMSE | Root-mean-square error |

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
