# Peer review of "Qualitative Numerical Analysis of a Free-Boundary Diffusive Logistic Model"

_mathematics, doi:10.3390/math11061296_

Round 1

Reviewer 1 Report

1. The original contributions need to be much better presented in the last paragraph of section \INTRODUCTION". All improvements, if they are, and new results must be described in this paragraph. Moreover, the
authors must explain about the objectives of next sections.
2. All acronyms should be defined.
3. The manuscript needs a revision in some places. In the follow I appoint some places in the text that I think
that need small corrections. Some improvements should be done from the point of view of English. They should
double check the mathematical formulations.
A professional proofreading revision is required.
4. In organizing the introduction, it is not good to only pile the related references together, it will be better to
give the relationship between these references
5. Please interpret the obtained results in the section Numerical result. It is important what you conclude to
them (It is very briefly)

6. Latest publications has to be added.

7. Plagiarism report is attached for improvement. 

Author Response

We thank and appreciate the referee’s suggestions and comments. All of them have been highly  taken into account in the new version of the manuscript. These general comments are used not only for this manuscript but for every one.

Reviewer 2 Report

The author of the paper studied the two-dimensional free boundary diffusion logistic model with radial symmetry, and obtained relatively good results, such as using finite difference method to solve the model, qualitative numerical analysis, numerical algorithm of spreading-vanishing dichotomy, etc. The results are satisfactory and meet the requirements for publication. Therefore, I recommend the paper to be published.

Author Response

We thank the reviewer for her/his analysis of our manuscript and comments.

Reviewer 3 Report

The field of the paper is relevant. There are comments.

1. Equation (2) was proposed by H. Hotelling in 1921 as a means to describe the growth and spread of a population. I think it's worth noting in the introduction with citing the paper 

Hotelling, H.,  1921,  A  Mathematical  Theory  of Migration, MA thesis presented at  the University of Washington; republished in 1978 in Environment  and  Planning  10:1223-1239.

2. The operator on the right side of equality (8) is called the Bessel operator. Equations and functional spaces with the Bessel operator were studied by I.A. Kipriyanov and his school. It would be good to refer to the pioneering work 

I. A. Kipriyanov "Fourier–Bessel transforms and imbedding theorems for weight classes",  Trudy Matematicheskogo Instituta imeni VA Steklova  V 89, 130-213 (1967).

and review

V. V. Katrakhov and S. M. Sitnik "The transmutation method and boundary-value problems for singular differential equations"  -- arXiv:1809.10887 [math.CA].

3.  For Theorem 5, it would be better to give a more detailed proof or give a link to the work where this proof can be found.

Author Response

  1. Equation (2) was proposed by H. Hotelling in 1921 as a means to describe the growth and spread of a population. I think it's worth noting in the introduction with citing the paper Hotelling, H., 1921, A Mathematical  Theory  of Migration, MA thesis presented at  the University of Washington; republished in 1978 in Environment  and  Planning  10:1223-1239.

The citation of the work of Hotelling suggested by the referee is now included in the Introduction and cited in the References Section.

  1. The operator on the right side of equality (8) is called the Bessel operator. Equations and functional spaces with the Bessel operator were studied by I.A. Kipriyanov and his school. It would be good to refer to the pioneering work I. A. Kipriyanov "Fourier–Bessel transforms and imbedding theorems for weight classes", Trudy Matematicheskogo Instituta imeni VA Steklova  V 89, 130-213 (1967), and review V. V. Katrakhov and S. M. Sitnik "The transmutation method and boundary-value problems for singular differential equations"  -- arXiv:1809.10887 [math.CA].

In the new version of the manuscript we have denoted the right side of equality (8) as Bessel operator and both works suggested by the referee have been included as references.

  1. For Theorem 5, it would be better to give a more detailed proof or give a link to the work where this proof can be found.

We have provided an additional clarification regarding Theorem 5 in our paper. It is important to note that the stability of the numerical solution is closely tied to the positivity condition established in Theorem 4. Theorem 5, therefore, serves as a natural extension of Theorem 4, where we prove that the numerical solution is indeed stable under certain conditions.

Reviewer 4 Report

Dear Colleagues,

In this article, the authors propose an efficient numerical method for the free boundary diffusive logistic model with radial symmetry. I think the article will be of interest for researchers involved in the study of biological invasions, epidemic spreading, wildfire propagation and other. One of the advantages of this article is that the authors proved important theorems for approval of numerical algorithm for spreading-vanishing boundary for general case of non-constant parameters. I have some comments on this article. I would like the authors to clarify in the article how the accuracy of solving IVP (11) by the Runge-Kutta-Fehlberg method effects on the accuracy of finding first positive root of f(x)? Algorithm 1 indicates that interpolate(u,x) is used. What the class of interpolation functions did use of the authors? The authors should clarify it. I recommend publishing the article "Qualitative numerical analysis of a free-boundary diffusive logistic model" when correcting thease questions.

Best regards, Reviewer.

Author Response

I would like the authors to clarify in the article how the accuracy of solving IVP (11) by the Runge-Kutta-Fehlberg method effects on the accuracy of finding first positive root of f(x)? Algorithm 1 indicates that interpolate(u,x) is used. What the class of interpolation functions did use of the authors? The authors should clarify it.

In the new version of the manuscript we explain how Runge-Kutta-Fehlberg method can provide highly accurate solutions to IVP (11). We comment that the interpolation method used is the cubic spline interpolation.

Reviewer 5 Report

In the revised  manuscript,  the authors  are proposed  an numerical algorithm  for a two-dimensional free-boundary diffusive logistic model with radial symmetry. Under the assumption about the radial symmetry, the original problem can be written as an one-dimensional free boundary PDE, which is solved by a combination of the front-fixing transformation with the explicit FDM. The numerical algorithm is found to be conditionally stable, consistent with the original PDE problem. Moreover, the scheme preserved positivity and monotonicity of the solution. 

Several important theorems have been proven. 

Theoretical statements have been illustrated by numerical examples. 

We have some comments:

1) In our opinion, the references  is incomplete. For example, it needs to add works by  authors:  Weiwei Ding, Rui Peng, Lei Wei, The diffusive logistic model with a free boundary in a heterogeneous time-periodic environment, http://dx.doi.org/10.1016/j.jde.2017.04.013.

The list goes on.

2) Proposed in the manuscript numerical algorithm  needs a comparative analysis with known numerical methods for solving a two-dimensional free-boundary diffusive logistic model with radial symmetry.

3) It is desirable for the authors to demonstrate the advantages of the proposed numerical algorithm using specific numerical examples.

In our opinion, the manuscript needs a major revision.

Author Response

  1. In our opinion, the references is incomplete. For example, it needs to add works by  authors:  Weiwei Ding, Rui Peng, Lei Wei, The diffusive logistic model with a free boundary in a heterogeneous time-periodic environment, http://dx.doi.org/10.1016/j.jde.2017.04.013.

The suggested work is now cited in the Introduction and included in the list of References. Other suitable references have been included (highlighted in red) in the Reference Section.

2. Proposed in the manuscript numerical algorithm needs a comparative analysis with known numerical methods for solving a two-dimensional free-boundary diffusive logistic model with radial symmetry.

We thank the referee’s suggestion and in the new version a comparison of our Front-Fixing Finite Difference method is compared with the well-known Level Set Method for solving free boundary PDE models. Now Table 1 illustrates this comparison for the 2D free-boundary diffusive logistic model with radial symmetry showing root mean square errors and CPU times of both methods. The spreading front of the population obtained by both methods is shown in the new Figure 8.

3. It is desirable for the authors to demonstrate the advantages of the proposed numerical algorithm using specific numerical examples.

By means of Example 3 we show in the new version the advantages of the proposed numerical method that guarantees conditional stability, positivity and monotonicity as it has been proved in Section 3. Also now in the Conclusions Section we comment the advantages of our proposed method.

Round 2

Reviewer 5 Report

In the revised version of the manuscript the  authors made changes according to the comments and recommendations.

The revised version of the manuscript can be published in journal.